# Association between Lameness and Indicators of Dairy Cow Welfare Based on Locomotion Scoring, Body and Hock Condition, Leg Hygiene and Lying Behavior

**DOI:** 10.3390/ani7110079

**Published:** 2017-11-05

**Authors:** Mohammed B. Sadiq, Siti Z. Ramanoon, Wan Mastura Shaik Mossadeq, Rozaihan Mansor, Sharifah Salmah Syed-Hussain

**Affiliations:** 1Department of Farm and Exotic Animal Medicine and Surgery, Faculty of Veterinary Medicine, Universiti Putra Malaysia, UPM Serdang, Selangor 43400, Malaysia; sadiquemohammed99@yahoo.com (M.B.S.); rozaihan@upm.edu.my (R.M.); 2Centre of Excellence (Ruminant), Faculty of Veterinary Medicine, Universiti Putra Malaysia, UPM Serdang, Selangor 43400, Malaysia; wmastura@upm.edu.my (W.M.S.M.); ssalmah@upm.edu.my (S.S.S.-H.); 3Department of Veterinary Preclinical Sciences, Faculty of Veterinary Medicine, Universiti Putra Malaysia, UPM Serdang, Selangor 43400, Malaysia; 4Department of Veterinary Clinical Studies, Faculty of Veterinary Medicine, Universiti Putra Malaysia, UPM Serdang, Selangor 43400, Malaysia

**Keywords:** animal welfare, body condition, lameness, locomotion, dairy cows, lying behaviour

## Abstract

**Simple Summary:**

Lameness is a major welfare issue in dairy cows. Locomotion scoring (LS) is mostly used in identifying lame cows based on gait and postural changes. However, lameness shares some important associations with body condition, hock condition, leg hygiene and behavioral changes such as lying behavior. These measures are considered animal-based indicators in assessing welfare in dairy cows. This review discusses lameness as a welfare problem, the use of LS, and the relationship with the aforementioned welfare assessment protocols. Such information could be useful in depicting the impact on cow welfare as well as in reducing the occurrence of lameness in dairy herds.

**Abstract:**

Dairy cow welfare is an important consideration for optimal production in the dairy industry. Lameness affects the welfare of dairy herds by limiting productivity. Whilst the application of LS systems helps in identifying lame cows, the technique meets with certain constraints, ranging from the detection of mild gait changes to on-farm practical applications. Recent studies have shown that certain animal-based measures considered in welfare assessment, such as body condition, hock condition and leg hygiene, are associated with lameness in dairy cows. Furthermore, behavioural changes inherent in lame cows, especially the comfort in resting and lying down, have been shown to be vital indicators of cow welfare. Highlighting the relationship between lameness and these welfare indicators could assist in better understanding their role, either as risk factors or as consequences of lameness. Nevertheless, since the conditions predisposing a cow to lameness are multifaceted, it is vital to cite the factors that could influence the on-farm practical application of such welfare indicators in lameness studies. This review begins with the welfare consequences of lameness by comparing normal and abnormal gait as well as the use of LS system in detecting lame cows. Animal-based measures related to cow welfare and links with changes in locomotion as employed in lameness research are discussed. Finally, alterations in lying behaviour are also presented as indicators of lameness with the corresponding welfare implication in lame cows.

## 1. Introduction

Intensive farming systems are now common practice to meet the increasing demand for milk in different parts of the world. This has led to the introduction of dairy cows to an environment arbitrarily different from the cows’ natural habitat, thereby triggering a range of welfare consequences. An animal is said to be in good welfare when it is able to express its innate behavior, free from distress and fear, in the absence of pain, and in good health [1]. However, these fundamentals of optimal welfare are often lacking with the advent of confining cows and persistent demands for high milk yield. As a result of these practices, outcomes such as chronic pain, discomfort, increased susceptibility to infectious disease and metabolic or physical fatigue are now common in dairy cows within intensive farming systems [2].

Lameness is a multifactorial condition and the most important welfare problem in dairy cows. Lameness is also regarded as a cause of economic loss owing to a reduction in milk yields, lowered reproductive performance and an increased risk of culling [3,4]. Farmers are often reported to underestimate the prevalence of lameness, thereby prompting a low perception of its impact on cow welfare, health and production [5,6]. With the rising occurrence of lameness in dairy herds globally, attempts to reduce the impact on welfare and production are needed. Locomotion scoring (LS) is widely used in detecting lame cows, in which gait properties are described to classify the severity on a numerical scale [7]. Events such as the small amount of time for farmers to observe lame cows, inadequate knowledge and inconsistencies in LS applications have encouraged the exploration of automated systems in lameness detection [8,9]. Nevertheless, the practical applications of the LS system on farms are limited. However, certain animal-based measures such as body condition scoring (BCS), hock condition and leg hygiene have been employed in assessing cow welfare, with recent findings suggesting vital associations with lameness. For instance, thin cows with low BCS (defined as BCS < 2 out of 5 scale) and poor hock condition have been reported to have a higher likelihood of becoming lame [10,11]. In some other studies, infectious causes of lameness and claw horn lesions were related to poor leg hygiene [12,13]. Amongst the behavioral alterations used in assessing cow welfare, resting or lying down activities have been described as potential indicators of lameness in dairy cows [14]. 

This review gives a brief introduction to lameness and gait changes that are used in detecting lame cows and the application of LS. The association between lameness and the aforementioned animal-based measures are discussed in relation to cow welfare. Lying behavioral changes and their potential role as indicators of lameness were also highlighted. In each section, factors that could influence the practical application of these measures as indicators of lameness and welfare were discussed.

## 2. Lameness in Dairy Cows

Lameness is a production-limiting disease and is regarded as the third most likely cause of the culling of dairy cows after mastitis and infertility [15]. Accordingly, lameness is an essential welfare problem, as studies have reported symptoms of distress and pain in affected dairy cows [16,17] as well as a negative impact on intrinsic behaviors such as lying down [18]. In lame cows, economic losses accrue in respect to reproductive performance. Milk yields might also be affected but remain undetected except where farm records are effectively monitored. Moreover, concerns such as under-diagnosis and effects on high-producing cows further complicate the problem of detecting ongoing loss [19,20]. The welfare implication is the likelihood for lame cows to be in pain, stress and unhealthy conditions in the herd without being detected. In addition, farmers’ awareness of the welfare implications of lameness problems have been generally reported to be low [6]. On occasions where farmers perceive lameness as a problem, another contributing factor to the underestimation of lameness was the adaptive behavior of cows to conceal signs of pain by restricting gait changes until the condition becomes severe [21]. In this regard, the search for techniques and indicators for a timely diagnosis of subclinical to clinical lameness in dairy herds becomes plausible. However, there is a need to present the definition of lameness and to understand normal locomotion performance in sound cows to appreciate any alteration.

### 2.1. Definition of Lameness 

Lameness is defined as the clinical presentation of impaired locomotion [22]. Olechnowicz and Jaskowski [23] described lameness as any condition characterized by alteration of gait caused by injury to the hoof or limb. However, a more elaborate definition is a resultant inclusion of the aforementioned features as the clinical manifestation of painful disorders, either as impaired mobility or abnormal gait and posture that are connected to problems in the locomotor system [21]. The degree of severity varies based on the type and location of the injury. Possible outcomes of the injury include stiff or asymmetrical limb movement to non-weight bearing presentation. Nevertheless, severe cases could result in lateral recumbency and increased lying down duration in the affected cow [18]. Hence, gait changes arising from pain and behavioral changes are important manifestations of lameness. 

### 2.2. Description of Sound Locomotion 

In order to identify lame cows, it is pertinent to understand the parameters that define a normal gait. Measures involving association between limbs, stride movement in footfall patterns and limb-to-claw movement have been used in describing animal gait. A stride incorporates three major features, which are walking, decisive steps and specific direction [21]. Hence, stride can be seen as a vector quantity based on its distance and directional components. In cows, stride ends up in the shortening of the limb and flexion of the joint when the hip, knee, hock and digital flexors are lifted above the ground. Philips et al. [24] divided strides as seen in cows’ locomotion into swing, support and suspension phases. The former entails the lifting of the limb above the ground, leading to gradual extension of the joints. At the supporting phase, the limb makes contact with the ground and a further exertion force on the solar area before the next swing phase. The suspension phase is the moment in which all the limbs make contact with the ground; hence, for a cow to walk, there cannot be a suspension phase with support provided by only two or three limbs. In addition, a normal locomotion, according to Hildebrand [25], should display less duration of support as compared to the swing time. Additional descriptions of gait include the duration of stance and swing phases during one stride [25] as well as the inclusion of time intervals between successive movements of the rear or fore limbs [26]. Telezhenko et al. [27] designated a spatial association between the limbs in the form of track-way diagrams. The system incorporated measures of movement rate such as stride length and tracking, coordination of the limbs and maintenance of balance. However, even in sound cows, there are certain cow level factors suggested to affect locomotion, such as lactation stage, age and cow height [21,27]. These factors need to be considered when applying any system of scoring to assess gait properties.

## 3. Signs of Lameness

### 3.1. Alteration in Gait Presentation

Various gait characteristics such as stride length, asymmetrical steps, speed, and weight distribution during the cows’ locomotion have all been employed in lameness detection. Accordingly, severely lame cows were reported to walk slower, and displayed a shorter stride and a reduced step angle and step length [27]. Flower et al. [28] demonstrated that, in addition to slow movement, lame cows exhibited longer stride times as weight distribution over the limbs was unequal compared to sound cows. In another study, lame cows displayed step overlap and negative tracking distance [29]. Step overlap is either a reduced or increased extension of stride between the limbs, where the hind limbs fail to be placed at the same position as the fore limbs immediately after the previous stride. In the same vein, lameness was depicted by increased abduction as seen in the lateral distance between the fore claw imprint and corresponding presentation of the rear claw [28,29]. Gait feature changes such as asymmetry in step length, width and time between the right and left limbs during locomotion were also reported in lame cows. In this context, Van Nuffel et al. [21] found that inconsistent gait manifested as a progression from initial swapping from short to normal strides before persistent shorter strides as the severity of lameness increases. 

### 3.2. Alterations in Posture and Presentation of Body Movements

Several postural changes are common in lame cows, including the presentation of the limbs when standing, back presentation and the position of specific parts of the body during locomotion. As shown in Figure 1, the two cows display the typical stance of a non-lame (right) and a lame cow (left). The presence of the hocked posture in the cow on the left is suggestive of lameness, as such a stance is adapted to relieve the pain present in the lateral claw [30]. Van der Tol et al. [31] showed that the lateral claw bears the majority of the body weight compared to the medial part during locomotion. Nevertheless, the hocked posture might be absent if either the fore limbs, medial claw or multiple claw lesions are present. The presentation of an arched back posture either at still or during locomotion has been associated with lameness in dairy cows [6]. The reason for such a posture was linked to the attempt to annul uneven weight distribution, depending on the limbs affected. Also, head bobbing—in the form of either nodding or vertical movement—in consonance with the moment the claws touch the ground was reported as a typical feature of lameness [28]. 

### 3.3. Alteration in Weight Bearing 

Non-lame cows normally display even weight distribution as a result of the balance between the claws and ground reaction force [32]. However, lame cows, in an attempt to reduce pain, redirect their body weight to the unaffected limbs [33]. Hence, while standing, the measurement of ground reaction force and weight bearing could be crucial in the assessment of lameness. According to Pastell et al. [8], more weight is often transferred to the healthy hind limbs if lameness occurs symmetrically in the front limbs. Conversely, weight is rarely directed to the front limbs when the cause of lameness is present in the hind limbs [8]. This was further established in later studies after quantifying weight distributions and leg weight ratios between sound and lame limbs [34]. Another important aspect is that weight bearing on unaffected limbs could also be induced as cows tend to kick. For a cow to kick, support needs to be provided by one rear limb, bearing most of the weight in the process. Chapinal et al. [34] and Chapinal and Tucker [35] found that lame cows exhibited increased step and kick behavior during milking compared to non-lame cows. However, there have been conflicting reports on the inclusion of the increased frequency of kick behavior as an indicator of lameness, as a similar event could be linked to presence of teat or udder injuries. 

## 4. Locomotion Scoring in Dairy Cows

Locomotion scoring (LS) is a useful assessment tool in the study, monitoring and prevention of lameness in dairy herds [36]. Locomotion scoring entails the observation of well-described gait and postural features as a cow walks on a flat surface. The five-point LS method developed by Sprecher [7] is one of the most frequently employed methods in lameness research. The presence or absence of an arched back is an essential feature assessed in the system [37].

Generally, the fundamental and consistent signs used in detecting lameness when applying LS include stride length, steps (asymmetrical gait), back presentation (presence of arched back), and the transfer of weight to the unaffected limbs, depending on the severity of lameness in dairy cows [38]. The first detailed LS system in cattle was described by Manson and Leaver [39] by using a nine-point scoring scale with specific features including tenderness, abduction and walking or rising ease. Subsequently, LS in cows was categorized into five classes by focusing on features such as gait asymmetry and locomotion difficulty [40]. The inclusion of head bobs as a gait indicator of lameness was made by Breuer et al. [41] prior to the modification by Flower and Weary [42] by introducing tracking up and joint flexion as additional measures. The LS system developed by the Welfare Quality^®^ Assessment Protocol for Cattle [43] entails the observation of irregular steps, the rhythm between successive claw placements and the different time of weight borne on each of the four feet. These aforementioned gait indicators were recently employed by Van Nuffel et al. [44] in categorizing cows into non-lame, mildly lame and severely lame. However, LS methods are mostly applied in free-stalls and are limited to the assessment of gait changes in response to pain. Ultimately, the diagnosis of lesions causing lameness requires the proper examination of the limb. In tie-stall herds, stance features such as the rotation of feet away from the body midline, foot resting, repetition of weight imbalance between limbs and uneven weight bearing during side movement are mostly assessed in lameness detection. According to Leach et al. [45], two or more of the listed gait indicators need to be present for a cow to be considered lame.

## 5. Challenges in the Application of LS

### 5.1. Reliability of LS Systems

There have been reports of certain weaknesses inherent in the use of LS, such as the difficulty in identifying cows at early stages of lameness and the undetailed description of the specific gait changes in affected animals [46,47]. A vital limitation is the subjectivity of the technique owing to intra-observer and inter-observer agreement and reliability [37]. Reliability is dependent on the quality and homogeneity of the sample population in a herd and the capability of observers to distinguish between lame and non-lame cows during LS [48]. Agreement, on the other hand, is the capability of observers to assign similar locomotion scores to sampled cows [49]. According to Schlageter-Tello et al. [37], the lack of a gold standard test and the degree of training among observers are major factors influencing agreement in the application of manual LS. However, the most widely used statistical measure of agreement in manual LS is the proportion of agreement (PA), where the acceptance threshold for good PA estimates is 75% [44]. Improvement in PA estimates was reported when LS scales were reduced from five to two levels comprising lame and non-lame [37,43]. According to Channon et al. [50], one of the reasons for variability is the unspecific description of the criteria for LS systems. In this context, observers might find it difficult to differentiate between moderately lame and mildly lame as seen in some LS systems. Horseman et al. [5] suggested a similar reason for the variability in prevalence of lameness estimates between farmers and veterinarians while applying the LS system. To improve the reliability of LS in lameness studies, factors that need to be considered with the potential of influencing locomotion performance include parity [51], walking surface [27], anatomical conformation [52], claw trimming [53], and degree of udder distension and lactation stage [9]. Other approaches include periodic retraining in order to reach acceptable levels of inter-observer reliability [46]. Authors have also reported improvements in the sensitivity of LS methods through the addition of certain gait indicators such as stride length, asymmetrical steps, tracking up [27], head bobbing, tracking up and joint flexion [42]. 

In addition to manual LS systems, automated systems involving computerized kinematic techniques, sensors and accelerometers [9,34,54] have been developed to detect lame cows and the presence of specific claw lesions. As reviewed by Schlageter et al. [37], these advanced systems have been reported with higher diagnostic values, with specificity (Sp) and sensitivity (Se) as high as ≥80% and 39–90%, respectively. However, the use of automated systems for detecting lame cows is limited, as the validation entails the use of the LS system as the gold standard. From the reports of diagnostic values ranging from 39 to 90% Se and 80% Sp in a few studies, the Sp value shows that automated systems are mostly accurate in detecting non-lame cows in contrast to truly affected cows. Nevertheless, a higher diagnostic value (Sp of 91.7%) was reported in a recent study where accelerometers and sensors were employed to detect slight lameness (LS of 2.5) by assessing standing bout and walking speed [55]. Overall, there are limited studies on the agreement and reliability of most automated systems for lameness detection as only few studies report the diagnostic properties. 

### 5.2. Behavioral Features and On-Farm Practical Applications

Over time, certain behavioral features in dairy cows have been shown to influence the reliability of LS systems. Cows have been recognised to hide notable signs of lameness in the presence of an observer as a behavior to evade predators [46]; gait changes might therefore only be presented when lameness is at an advanced level. Also, individual cows could adapt differently to potential causes of lameness, and therefore gait changes might be the product of the animal’s capacity to withstand the ongoing pain. Flower et al. [9] showed that cows walked more soundly with longer strides after milking than before milking; the social behaviors of the cows or the reduced distension of the udder were the suggested reasons for such behavior. 

For practical application of LS, the aim is to assign scores to cows as they move undisturbed on a flat, non-slippery surface for a considerable duration to assess multiple strides. However, in most farms, provisions to accommodate the aforementioned criteria are often lacking. The presence of manure and floor designs might also influence the frictional and compressional forces that mediate mobility [31,44]. Stall designs might also limit the chance for observation of multiple strides. Accordingly, Flower et al. [28] reported a high variability (76%) in outcomes when only short strides were captured in assessing lame cows. Another factor that influences the practical use of either visual LS or automated systems is the farmers’ preference. Van De Gucht et al. [56] reported that farmers who attach more importance to lameness are more willing to accept the use of automated systems, while visual LS was more preferred by all famers. 

## 6. Association between Walking Surfaces Types and Locomotion Performance 

Housing design is vital for the maintenance of good welfare in dairy cows. Floor type and its influence on locomotion performance in dairy cattle were first suggested by Albright in 1997. Subsequently, floor features such as abrasiveness and hardness leading to insufficient friction and traction—as present in concrete floors (CF)—were suggested to negatively impact the claw health and locomotion of dairy cows [31,57]. In this context, the use of cushioning floor surfaces, such as through the use of rubber flooring (RF), has been reported to improve gait properties. This includes reduced muscular activity in the hind limbs [58] and similar stride length when compared to locomotion on pasture [59]. 

Additionally, the influence of floor types on lameness occurrence has been reported extensively. This has been linked to prolonged standing and walking on hard or abrasive surfaces leading to sub-optimal claw health including claw horn lesions (CHL). Fjeldaas et al. [60] reported that the risk of higher LS was three times higher in cows on CF compared to RF. In a study evaluating locomotion performance in dairy herds on CF and straw yards, a total of 1% and 46% of all the observed gaits in cows on straw yards and those from the cubicle housed group, respectively, were scored as lame [61]. Similarly, following the comparison of locomotion in lame and non-lame cows on RF and CF, Telezhenko et al. [27] showed that the moderately lame cows walked with a significantly wider posture on the CF than RF, while a similar group on CF had a smaller step angle compared to the non-lame counterparts. However, there was no significant difference between non-lame cows and lame cows on RF [27]. Specifically, cows affected with digital dermatitis (DD) in a study on straw yard walked significantly better than the same group on CF, and about 81% and 1% of all observed gaits on a straw yard were scored as normal and clinically lame, respectively. However, 46% and 27% of dairy cows were scored as normal and lame on CF [62]. This suggests that differences exist in locomotion performance between lame and non-lame cows, and within lame cows, on various flooring systems. Rubber flooring offers better comfort to a cow’s hoof. Invariably, improving mobility might mask the presence of claw lesions, especially at the subclinical stage where detectable changes in locomotion are absent. 

## 7. Body Condition Scoring and Association with Lameness

Body condition scoring (BCS) has been described as a technique for assessing the condition of livestock at particular periods to achieve equilibrium between economic feeding, yield and adequate welfare [63]. The BCS technique is a manual assessment with a corresponding outcome that is recorded on a numerical scale as thin, good or grossly fat. Leach et al. [64] explained that the inclusion of body condition in evaluating welfare is to identify animals that are too thin or too fat, since body reserves in both cases are linked to increased likelihood of disease. The association between body condition and lameness has been studied extensively. Lame cows are believed to lose body condition class over time due to changes in feeding habits or intrinsic pain affecting feed conversion [65]. Recent findings have shown that cows with low body condition are more likely to become lame [11,66].

In relation to cow welfare, the association between BCS and lameness has been studied by considering the effect on measures of productivity. In a study, the BCS changes between cows with and without claw horn lesions (CHL) and their corresponding conception rate showed that cows with good BCS without CHL produced more milk and were more likely to conceive than those with low (thin) and high (fat) BCS with and without CHL [67]. Similarly, on a more widespread BCS scale (1–5), cows with BCS < 2.5 (thin) were associated with an increased risk of lameness in the subsequent zero to two months for all cases of lameness and two to four months for claw horn lesions (sole ulcer and white line disease) [68]. Accordingly, an important structure within the claw capsule that has been established to play a crucial role in the development of CHL is the digital cushion (DC) or fatty pad [69]. The DC serves as a shock absorber to the pedal bone (3rd phalanx) which bears most of the weight of the cow at the claw-floor interface. However, the pedal bone becomes unstable at the peri-parturient period due to hormonal changes, thereby predisposing the internal capsule to displacement injuries [69]. Also, the DC is not well developed in first heifers until the second and third lactations and is often depleted in thin cows with low BCS [70]. 

Lame cows affected with CHL have been characterized with thin DC, suggesting that the cows might have been in a low body condition prior to the onset of lameness, as the protective function of the DC to the sole and white line was compromised [11]. Randall et al. [11] reported that a low BCS at a specific period of 8–16 weeks was associated with an increased risk of lameness before repeated lameness, and three weeks of low BCS before the first lameness was noted. Similarly, findings from a study highlighted the importance of maintaining cows in good BCS to minimize the risk of developing CHL [67]. However, recent findings have suggested that the thinness of the sole tissue does not necessarily arise from the depletion of body fats and DC, but could also be due to other factors such as the integrity of the suspensory apparatus, calving, herd, and lesion presence [66].

In contrast, a study reported increased odds of lameness in cows with high BCS (≥4.25) [71]. Nevertheless, the basis for such an association needs to be more thoroughly investigated. Some authors have related the event to increased weight in the pelvic region that could be transferred to the hind limbs, thereby causing overload. Another likely pathway is the nutritional changes in fat cows as they approach calving, due to reduced appetite and low fibre intake, thereby increasing the susceptibility to ruminal acidosis and onset of laminitis [72]. In one study, European dairy cows affected with subclinical ketosis were reported to have increased odds of lameness [73]. 

## 8. Hock Condition and Lameness Occurrence

Although claw lesions remain among the major causes of lameness in dairy cows [74], hock lesions and injuries are becoming a persistent problem in intensively managed dairy farms [75]. The term “hock lesion” is used to describe various anomalies such as hair loss, visible wounds, broken skin, and localized and general swelling of the hock [76]. In dairy cows, the absence of fatty tissues and muscles around the hock makes the region prone to trauma and damage to the skin. Consequently, the development of hock lesions is directly influenced by the nature of the lying surface of hard and abrasive [77]. In welfare assessment, the lateral aspect of the hock is often examined and suggested to be the most affected area. Poor hock conditions are often manifested as hair loss, swelling or ulceration [78]. 

The hock condition score (HCS) measures the severity of hock lesions on various scoring scales based on features ranging from normal to substantial injuries. The assessment is important in free-stalls and loose cubicle housing, as such provisions encourage movement and interaction with stall designs. One of the simplest hock scoring systems was described by Rutherford et al. [79] by using a two-scale scoring system divided into (1) no skin damage and (2) damaged skin with various levels. The advantages of such an HCS system is the repeatability and reduced inter- and intra-observer variability of the results, as found in the use of lower scoring scales in LS. However, the system lacks the ability to capture several manifestations of hock injuries. Selected HCS methods employed in assessment of cow welfare and their clinical descriptions are presented in Table 1.

Several studies have demonstrated the inter-relationship between occurrence of hock lesions and lameness in dairy cows. An earlier study in the United States (USA) by Whay [47], suggested that 80% of the 53 dairy farms investigated needed to reduce hock lesions in order to minimize the incidence of lameness. By investigating the factors associated with hock lesions, a higher incidence was reported in inorganic herds (49.7%) and free-stalls (46.0%) as compared to organic herds (37.2%) and straw yards (25%) [79]. Nevertheless, housing cows in free-stalls with less access to pasture grazing was previously reported to increase the incidence of claw lesions causing lameness [2,85]. Such housing conditions might favour the occurrence of hock injuries and lameness. Chapinal et al. [34] found a positive correlation between lameness and hock injuries and suggested that reporting and monitoring the prevalence of both conditions could assist in improving cow welfare in dairy herds. Several authors have reported similar findings by showing that hock lesions ranging from hair loss to severe ulcers are associated with higher locomotion scores (LS > 3) and lameness occurrence (Table 2). 

As highlighted previously for the other welfare assessment systems, certain environmental factors could influence the association between hock condition and lameness occurrence. For instance, the level of comfort from the lying surface might influence the severity of hock lesions as well as increase the risk of lameness [86]. Hence, the pathogenesis of hock lesions and the direction of the event as related to lameness need to be investigated. Severe hock lesions could initiate painful sensations leading to lameness, while a prolonged duration of lying down in lame cows on hard and abrasive surfaces might precipitate hock injuries. Another aspect that might contribute to the occurrence of severe hock injury is floor slipperiness. A notable technique for assessing the slippery index of floors in dairy housing was developed by Grandin [87] based on the frequency of slips and falls within a specific period. A recent study reported higher odds (Odds ratio, OR = 2.0) of cows being lame and with hock lesions (OR = 1.4) when reared on slippery floors compared to non-slippery floors [13]. Telezhenko et al. [88], in a recent study involving gait analysis and skid resistance of different flooring systems in dairy housing, showed that rubber mats had the highest coefficient of friction and skid resistance values compared to concrete and mastic asphalt floors. This further depicts lower slipping tendencies in cows when housed on rubber mats or floors. Overall, the aforementioned events show that preventive measures for hock lesions have the potential of reducing lameness incidence, contributing to general improvements in cow welfare.

## 9. Leg Hygiene Score and Lameness Occurrence 

Cleanliness is a significant aspect of animal welfare, through links with lameness and mastitis. In the assessment of cow welfare, Napolitano et al. [93] included the genital area, back of the udder, and lower part of the hind limbs for scoring cow cleanliness, also known as the cow hygiene score. Cook [94] described the leg cleanliness scoring system based on the level of manure contamination of the lateral aspect of the lower hind legs. Recent findings by Solano et al. [13] indicated that the assessment of leg cleanliness could enhance the understanding of the association between lameness and herd cleanliness. However, since most of the studies were cross-sectional, the role of leg hygiene either as a risk factor for lameness or consequent of lameness needs to be further investigated. 

Rodriquez-Lainz et al. [95] first pointed out that infectious causes of lameness, such as digital dermatitis (DD), could be attributed to unhygienic environments that enhance the growth of pathogenic organisms capable of invading the digital skin. DD is of greater significance in confined cows, especially in free-stalls where exposure to manure slurry is persistent, thereby predisposing cows to poor leg hygiene scores [96]. DD was also described as a lameness condition potentiated by unhygienic environments, dirtier herds and persistent exposure of the hooves to contagious agents [97,98]. Generally, the pathogenesis of DD and the role of leg hygiene are still investigated based on changes in claw traits either as risk factors for the development of DD or the resultant outcome of an ongoing problem [53]. However, authors have reported a positive relationship between leg cleanliness and prevalence of DD. Cows that had predominantly dirtier legs demonstrated higher risk (OR = 2.44) of developing DD [80]. Solano et al. [99] also found that poor leg cleanliness at cow level was associated with higher prevalence of active lesions of DD. 

In agreement with the multifactorial nature of lameness, environmental factors such as floor designs could influence hygiene and the risk of infectious claw lesions. In this context, grooved CF is likely to retain a more sufficient amount of manure slurry than textured CF following scraping. Hence, grooved CF was identified as a risk for high prevalence of DD in some UK dairy herds [100]. Similarly, cows on slatted floors with a manure scraper had lower odds of developing interdigital horn erosion than those on standard slatted floors. In addition to manure slurry, damp conditions leading to exposure of the cows’ feet to moisture also increases the risk of DD [101].

## 10. Association between Lameness and Lying Behavior

### 10.1. Importance of Lying Behavior and Lameness Occurrence

The ability of an animal either in its natural or artificially produced habitat to exhibit its natural behavior is of great welfare importance [1]. Accordingly, lying down is a behavioural need for the dairy cow. Ideally, a cow lies down for about 12–14 h per day and sleeps for 30 min within the stated timeframe. The importance of lying down ranges from adequate resting of the animal, efficient rumination, greater space for other cows’ movement, and maintenance of claw health by drying off [102]. Additionally, a study reported increased blood flow to the mammary gland by 30% when cows lie down, thereby leading to higher milk yields [103]. Moreover, the duration allocated to resting (12–14 h/day) gives an insight into the significance of the natural behavior to the well-being of the cow.

Lying and resting is an indication of welfare, and studies have suggested several ways of quantifying these behaviors. They include the ease of performing the activity [104], total lying time, number of lying bouts, duration of lying-down and getting up sequences [105]. Deviations from the budgeted time for lying down affect the allocated time for feeding and standing in the form of compensatory reaction [106]. A notable outcome is longer standing time, which could contribute to the development of claw lesions, especially on hard, wet and abrasive surfaces [107]. Studies have revealed the variation in lying time between lame and non-lame cows. In this regard, lame cows lie down for about 38 min to 0.6 h/day longer, and also with longer bouts [14,108]. In addition, high LS was reportedly associated with increased lying time and frequent bouts [109]. Few studies have demonstrated the impact of specific claw lesions on lying behavior. Lame cows affected with severe DD were observed to have spent more time lying down on CF compared to straw yards [61]. Lying down time was also reported to be highest in cows affected with DD, followed by sole ulcers [110]. In addition to being an indicator of lameness, lying behavioral changes could also be applied in assessing the risk of lameness. 

Consequently, automated systems used for measuring lying time with the ability to detect mild changes in lying behaviour have been employed in lameness studies [109]. In a recent study, cows presented with longer lying times and higher duration of bouts with 3.7 and 1.7 increased odds of being lame respectively compared to non-lame cows [105]. Necharitzky et al. [18] also reported that lame cows affected with claw horn lesions laid down significantly longer than healthy cows.

### 10.2. Practical Applications and Limitations of Lying Behavior Assessment

Well-known environmental factors that have been evaluated in association with lying behavior and lameness are bedding and stall designs. Dairy cows were reported to prefer lying down on softer surfaces, irrespective of the conditions of the limbs [111]. A study pointed out higher incidence of clinical lameness in cows on rubber mats (24%) compared to those on sand (11.7%) in a confined dairy herd. Also, the same study reported longer standing time in non-lame cows on rubber mats, thereby indicating discomfort in lying down compared to sand [105]. Similarly, in non-lame cows, lying down duration was greatest on rubber mats compared to sand and concrete [112]. Therefore, applications of lying behavior in lameness studies need to take into consideration factors such as comfort, stall designs, milking patterns and feeding management, as they could influence normal locomotion. 

The direction of the relationship between lying behavior and lameness needs to be further elucidated. There are indications that changes in lameness might induce changes in lying behavior, or the other way around [44]. In addition, there are other conditions that induce longer lying times and bouts in cows aside from lameness issues. Hence, the assessment of lying behavior might be useful as a tool for further examination of the cow in a similar manner to the LS system. This might entail routine claw assessment for the presence of ongoing lesions or injuries. There also seems to be a complex relationship in assessing the welfare implications of lameness on lying behavior in dairy cows. Higher duration of lying down in lame cows might predispose the hock area to infection depending on the hygiene of the lying surface and overall herd cleanliness. Furthermore, the proximity of the udder to a lying surface with persistent exposure to manure might contribute to secondary infections such as mastitis [113]. However, more research is needed to arrive at the relationship and direction of the event. 

## 11. Conclusions 

The application of LS for the identification of lame cows requires well-defined criteria of gait features to improve the reproducibility of results. Similarly, the practical applications are limited by the availability of adequate farm facilities to ensure accurate outcomes. However, the other welfare assessment protocols discussed herein are associated with lameness as a potential risk factor at cow level. Lying behavioral changes are also a potential indicator of lameness in the free-stall system. A better understanding and demonstration of the relationship between lameness and the assessment scoring systems could enhance farmers’ awareness in appreciating the welfare implications of lame cows and promote the provision of good welfare. 

## Figures and Tables

**Figure 1 animals-07-00079-f001:**
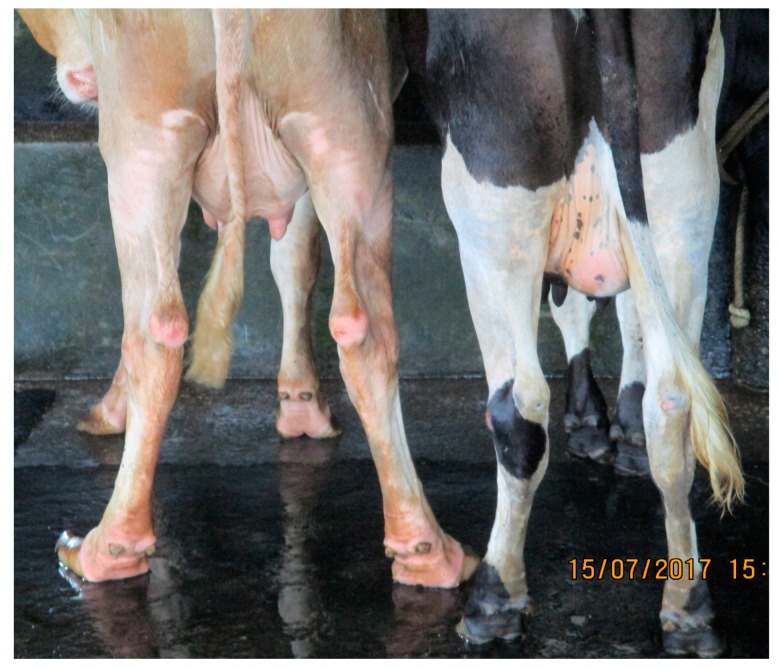
Stance of a non-lame cow (**right**), and a lame cow (**left**) displaying cow hock posture (Posterior view).

**Table 1 animals-07-00079-t001:** Commonly-used hock condition scoring (HCS) systems and the criteria employed.

Reference	HCS Scale	Clinical Criteria and Description of the Hock Condition					
		Normal	Hair Loss	Skin Changes	Ulceration	Swelling	Distinct Features
Rutherford et al. [79]	1–2	√	√	√	-	√	Absence of scores to categorize different level of clinical manifestations, hock ulceration not mentioned
Lombard et al. [80]	1–3	√	√	-	-	√	Hock ulceration and skin changes are not considered
Ahrens et al. [81]	0–4	√	√	√	-	-	Well description of skin changes but no consideration for ulceration and swelling hock
Potterton et al. [78]	0–3 for each category	√	√	-	√	√	Well description of different levels of hair loss, hock ulceration and swelling. Skin changes were not considered
Lobeck et al. [82], Gibbons et al. [83]	1–3	√	√	-	-	√	A single score for the each category
Van Gastelen [84]	0–3	√	-	-	-	√	Only used presence of lesion and swelling as manifestations of hock injuries

Note: √ depicts if criteria is included in the scoring system.

**Table 2 animals-07-00079-t002:** Selected studies and findings involving the association between lameness and hock lesions in dairy cows.

Reference	Housing Type	Number of Farms and Location	Association between Hock Lesions and Lameness	Other Findings
Zurbrigg et al. [12]	Tie stall herds	317 farms in Ontario, Canada	Prevalence of hock lesions and lameness based on ached back and rotation of hind claw were 44%, 3.2% and 23%, respectively	Faulty design of stall dimensions
Nash et al. [89]	Tie stall herds	100 farms in Ontario (*n* = 60) and Quebec (*n* = 40), Canada	Mean prevalence of hock lesions in cows was 58 ± 18% and increased odds of hock lesions in lame cows	Stall design are not in accordance with recommendations
Bouffard et al. [90]	Same as above	100 farms in Canada	Prevalence of lameness and hock lesions were 25% and 58% respectively	
Richert et al. 2013 [91]	Organic and small conventional farms	292 farms in United States	Correlation between prevalence and hock lesions prevalence with suggestion that similar risk factors influence both conditions	
Brenninkmeyer et al. [86]	Cubicle dairy designs	105 farms in Germany and Austria	High mean prevalence of hock lesions (50%; range 0–100%), and correlation between lameness and hock lesions prevalence at animal and herd level	
Adams et al. [92]	Free-stalls	191 dairy operations in the USA	Prevalence of mild (LS = 2) and severe lameness (LS = 3) were 6.9% and 2.6% respectively while prevalence of mild (score 2) and severe hock lesions (score 3) were 10.1% and 2.6% respectively	Sand bedding and access to pasture improved LS and hock conditions
Solano et al. [13]	Free-stalls	141 dairy farms in Canada	Increased odds (OR = 1.4) of lameness in cows with injured hocks compared to cows with normal hock condition	
Chapinal et al. [34]	Free-stalls	34 farms in China	Mean prevalence of clinical and severe lameness were 31 ± 12 (7–51) and 10 ± 6% (0–27%) respectively. Prevalence of minor and severe hock lesions were 40 ± 20 (6–95) and 5 ± 9% (0–50%) respectively	Deep bedding decreased the prevalence of all hock lesions

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
