# Peer review of "Association between Lameness and Indicators of Dairy Cow Welfare Based on Locomotion Scoring, Body and Hock Condition, Leg Hygiene and Lying Behavior"

_animals, 2017, doi:10.3390/ani7110079_

Round 1

Reviewer 1 Report

Line

Comment

4

for what?

.. as respective indicators

48

it is not   applicable...

a certain   connection is present, but not on a linear scale

78

This is a   excellent review article but I would prefer to add a chapter about literature   review. A review is always also a meta analysis. How were the literature   accessed? Just convenient sampling or systematic review in online databases   with specific keywords. 

159

insert: (right)

159

insert: (left)

299

review the sentence

336

Subclinical ketosis   is also mentioned at this stage as a risk factor for lameness

474

The implication of feeding, i.e. minerals, ketosis, NEFA should be mentioned at least as   additional risk factors with a review reference

The influence of   claw trimming should also be mentioned by a sentence and a reference

487

University?

Author Response

Dear Reviewer 1,

please find the response to your review in the attached file.

Thank you.

Best regards,

Siti Ramanoon 

Reviewer 2 Report

The focus of this paper is not very clear.  The Introduction is based on the idea that detection of lameness can be improved by using BCS, hock lession, lying behaviours, etc., but there is not any evidence of this provided in the manuscript. Instead the authors seem to confuse associations with causation, which severely limits their ability to formulate a coherent narrative.  Furthermore, there is not any real data presented to support the general idea that the inclusion of these factors are required to improve the detection of specific cows who are lame.  Finally, this paper would benefit from assistance with language.  There are numerous grammatical errors.  It is also unnecessarily, which reduces the readability of the manuscript and distracts from their argument.

Author Response

Dear Reviewer 2,

please find the attached file for the response to your review.

Thank you very much.

Best regards,

Siti Ramanoon
